# The Background K^+^ Channel TRESK in Sensory Physiology and Pain

**DOI:** 10.3390/ijms21155206

**Published:** 2020-07-23

**Authors:** Alba Andres-Bilbe, Aida Castellanos, Anna Pujol-Coma, Gerard Callejo, Nuria Comes, Xavier Gasull

**Affiliations:** 1Neurophysiology Laboratory, Department of Biomedicine, Medical School, Institute of Neurosciences, University of Barcelona, Casanova 143, 08036 Barcelona, Spain; albaandres@ub.edu (A.A.-B.); aida.castellanos@gmail.com (A.C.); annapujolcoma@gmail.com (A.P.-C.); gerard.callejo@ub.edu (G.C.); nuriacomes@ub.edu (N.C.); 2Institut d’Investigacions Biomèdiques August Pi i Sunyer (IDIBAPS), Villarroel 170, 08036 Barcelona, Spain

**Keywords:** potassium channel, pain, sensory neuron, somatosensation, migraine, hyperexcitability

## Abstract

TRESK belongs to the K_2P_ family of potassium channels, also known as background or leak potassium channels due to their biophysical properties and their role regulating membrane potential of cells. Several studies to date have highlighted the role of TRESK in regulating the excitability of specific subtypes of sensory neurons. These findings suggest TRESK could be involved in pain sensitivity. Here, we review the different evidence available that involves the channel in pain and sensory perception, from studies knocking out the channel or overexpressing it to identified mutations that link the channel to migraine pain. In addition, the therapeutic possibilities are discussed, as targeting the channel seems an interesting therapeutic approach to reduce nociceptor activation and to decrease pain.

## 1. Introduction

Potassium channels integrate the largest family of ion channels, with about 80 genes encoding for alpha subunits [1]. Four large families of K^+^ channels have been identified in mammals according to their structural and functional characteristics: voltage-gated K^+^ (K_v_) channels, Ca^2+^-activated K^+^ (K_Ca_) channels, inwardly rectifying K^+^ (K_ir_) channels, and two-pore domain background K^+^ (K_2P_) channels. They participate in numerous cellular functions in almost all cell types of the body, including K^+^ homeostasis, cell volume regulation, cellular excitability, and proliferation. In particular, the polarity of the plasma membrane depends on the function of specific K^+^ channels that allow the K^+^ efflux through the membrane according to its electrochemical gradient, driving the membrane potential of the cell to negative values (near −80 mV). This value is close to the equilibrium potential for K^+^ due to the high concentration of K^+^ inside the cells (about 140 mM) compared to extracellular medium (4 mM). In addition to their role in setting the resting membrane potential, K^+^ channels regulate cellular excitability. This is especially important in neurons, where potassium currents oppose the depolarization caused by other excitatory Na^+^, Ca^2+^ or nonselective cation channels, thus contributing to regulate neuronal firing and shaping the action potential.

Among the different K^+^ channel families, the family of K_2P_ K^+^ channels was the last one identified and described, which, to date, has 15 members to date, grouped into six subfamilies (TWIK, TREK, TASK, TALK, THIK, and TRESK) based on sequence and functional similarities [1,2].

The first K_2P_ channel identified was TWIK1, for Tandem of pore domains in a Weak Inward-rectifying K^+^ channel [3]. Now, this subfamily also contains TWIK2 and KCNK7 (K_2P_7.1). The TREK (TWIK-RElated K^+^ channel) subfamily contains TREK1, TREK2, and TRAAK (TWIK-Related Arachidonic acid Activated K^+^) channels. Members of this subfamily are activated by arachidonic acid, polyunsaturated fatty acids (PUFAs), volatile anesthetics, and pain-related stimuli. The TASK (TWIK-related Acid-Sensitive K^+^ channel) subfamily contains TASK1, TASK3, and TASK5 (K_2P_15.1, KCNK15), and these channels have the common property of being inhibited by extracellular acidification. The TALK (TWIK-related ALkaline pH-activated channel) subfamily includes TALK1, TALK2 (K_2P_17.1, KCNK17) and TASK2 that have an important sensitivity to extracellular alkaline pH. The THIK (Tandem pore domain Halothane-Inhibited K^+^) channel subfamily is composed by THIK1 and THIK2, and both channels are inhibited by halothane. Finally, the TRESK (TWIK-RElated Spinal cord K^+^) subfamily has only one member, TRESK (K_2P_18, KCNK18), which has the lowest structural and functional similarity to other K_2P_ channels, and it is the only K_2P_ regulated by intracellular Ca^2+^ concentration through calcineurin-mediated dephosphorylation.

In contrast to other K^+^ channel families that have one pore-forming domain for each subunit, K_2P_ channels have two pore domains and four transmembrane domains. Consequently, functional K_2P_ channels are dimers and not tetramers as most of the other K^+^ channels are. K_2P_ channel activity produces constitutive leak currents, which are mostly independent of the membrane potential, with some minor exceptions. As mentioned earlier, their main role in most cell types is attributed to the regulation of membrane potential, as they constitute the leak of potassium through the plasma membrane (accordingly they are commonly refereed as leak or background potassium channels) that equilibrates with the function of the Na^+^/K^+^ pump and help to set the resting membrane potential. For this reason, they influence neuronal excitability over a wide range of membrane potentials, especially between resting and action potential threshold. Thus, K_2P_ channels are one of the main sustained K^+^ conductance that establish the resting membrane potential in neurons, where they can also shape the duration, frequency, and amplitude of the action potential.

Basic biophysical properties of this family of channels, regulation, and interaction with other proteins are reviewed elsewhere, including some comprehensive and extensive reviews [2,4,5,6,7]. This review is intended to highlight the role of the last described K_2P_ member, TRESK, in somatosensory perception and pain. The role of other K^+^ channels in pain, including K_2P_s, has been nicely reviewed in several articles [8,9,10,11].

## 2. TRESK Identification and Expression

TRESK was initially identified and cloned from human spinal cord in 2003 [12]. Later and despite its low homology (65%), the mouse ortholog (initially identified as TRESK-2) was cloned from cerebellum and testis by other groups [13,14]. Initially, TRESK was only reported to be expressed in spinal cord in humans, but, in mice and rats, it was also detected in spleen, thymus, pancreas, kidney and testis [14], although other studies have not found a significant expression on some of these organs [15]. In the rodent nervous system, TRESK expression has been detected in cortex, cerebellum, brainstem, and spinal cord, but, especially, a high expression of the channel was found in sensory neurons of the dorsal root (DRG) and trigeminal ganglia (TG), as well as in neurons of the superior cervical ganglion and nodose ganglion [15,16,17,18,19,20,21]. Within sensory ganglia, TRESK seems predominantly expressed in small and medium diameter sensory neurons that mainly correspond to nociceptive neurons [16,22,23,24,25].

Recent studies of massive RNA sequencing, both bulk or at a single-cell level, have identified TRESK expression in selected subpopulations of mouse and human DRG sensory neurons [26,27,28,29,30]. In addition, trigeminal sensory neurons show a high expression of TRESK with a similar distribution pattern [31,32,33,34]. In particular, TRESK shows a predominant expression in non-peptidergic IB4^+^ sensory neurons (NP), which confirms previous immunohistochemistry data. NP1 and NP2 subgroups of non-peptidergic neurons show an important expression, while peptidergic sensory neurons (PEP2) show little expression [27]. The characteristic gene expression pattern of these subgroups (NP1, 2) suggests their involvement in neuropathic pain and itch. In addition, a significant expression of TRESK was detected in a subpopulation of low-threshold mechanoreceptors (NF1) expressing high levels of TrkB and likely involved in the detection of touch and propioception [26,27]. A similar TRESK expression pattern has been reported in subsets of sensory neurons from the trigeminal ganglia [32]. Recent studies using labeled subpopulations of DRG neurons from genetically modified mice have also identified TRESK expression in MrgprD^+^ non-peptidergic nociceptors, in CGRP^+^ peptidergic nociceptors and in Aδ-low threshold mechanoreceptors (TrkB^+^) [35]. At a functional level, native single channel currents with electrophysiological and pharmacological properties of TRESK have been reported in mouse and rat sensory neurons, corroborating expression studies [16,19,36,37]. TRESK, together with TREK-2, seems to constitute the major component of K^+^ background current in medium- and small-sized sensory neurons [16].

## 3. Involvement of TRESK in Pain

From its initial identification and by analogy to other channels of the family, such as TREK-1/2 or TRAAK, a putative involvement of TRESK in pain sensitivity was suggested. As mentioned earlier, TRESK gene expression is elevated in sensory neurons, specifically in small and medium-sized neurons that mainly correspond to nociceptors. In addition, when compared to other channels of the K_2P_ family, TRESK presents the highest expression in rodent sensory neurons, followed by TRAAK, TREK2, TREK1, TWIK1, and TWIK2, although relative expression may vary between studies [19,21,25]. In humans, the most expressed channels in DRG and TG are THIK-2, TASK1, and TWIK1, followed by TREK1, TASK2, and TRESK [33], but other studies found TRESK as the most expressed channel in human TG [20,31,38].

### 3.1. TRESK Expression and Sensory Neuron Excitability

Studies using mice with a total or a functional knockout of the channel have shown that TRESK ablation enhances the excitability of nociceptive sensory neurons [19,23,25,39], confirming that the channel regulates neuronal excitability. This is accompanied by a decrease in total K^+^ current and an increase in membrane input resistance. All this data indicates that a decrease in channel expression or function facilitates neuronal depolarization and activation; thus, in physiological conditions, the channel acts as a brake to prevent neuronal activation in response to depolarizing stimuli.

In this sense, a reduction of TRESK expression has been reported in different models characterized by hyperexcitability of sensory neurons. In an initial report, Tulleuda el al. described a marked decrease in TRESK expression in rat lumbar DRGs three weeks after sciatic nerve axotomy [18]. In addition, a reduction of TRESK expression by siRNA injection (thus mimicking the effects of axotomy) resulted in an enhanced sensitivity to mechanical stimuli but not to radiant heat stimulation. The study showed that, similarly to other ion channels and particularly some K^+^ channels [10], peripheral injury or axotomy reduces the expression of TRESK, consequently enhancing sensory neuron excitability. Other studies reported similar effects using different models of chronic neuropathic or inflammatory pain [21,40,41]. Interestingly and despite the fact that sensory neuron primary culture is a common model used in many studies, ganglia dissociation produces an inevitably axotomy where neurons become injured. Consequently, they show an enhanced expression of known injury markers (e.g., ATF3 or Cacna2d1) and quickly become hyperexcitable. Similar to what occurs after axotomy, a decrease in TRESK expression was observed after culturing DRGs [18]. In fact, a recent report that has compared the transcriptome of native versus cultured DRGs shows that the latter display an increased expression of several genes associated with nerve injury or inflammation (BDNF, MMP9, ATF3, or GAL) which indicates an injury-like phenotype of cultured DRGs [42]. Again, TRESK was downregulated in cultured neurons compared to its native expression in the ganglion. Whether expression of the channel recovers its normal levels after injury or if reinnervation occurs it is still unknown.

Regardless of possible effects of cell culture, studies comparing sensory neurons from WT or TRESK KO mice have found significant differences in their excitability. Weir et al. have shown that deletion of TRESK enhances the excitability of non-peptidergic (IB4^+^) trigeminal sensory neurons while the excitability of peptidergic neurons was not significantly modified compared to WT cells [23], consistent with the higher expression of the channel in the non-peptidergic neuron population. We reported a similar effect on DRG nociceptive neurons from a different TRESK KO mice, although in this case, no distinction was made between peptidergic and non-peptidergic neurons [25]. A third study reported similar results in TG neurons but, surprisingly, lumbar DRG neurons from the same TRESK KO mice were not found to be hyperexcitable [39]. In a functional TRESK[G339R] knockout mice, sensory neurons also showed a reduced current threshold to fire an action potential and an enhanced excitability [19,24]. Surprisingly, none of these studies found significant differences in resting membrane potential after deleting TRESK, suggesting that either TRESK removal is compensated by the concourse of other channels or that the channel has a minor role on setting the resting membrane potential. Besides, all these studies indicate that TRESK have a major impact on the range between the resting membrane potential and the action potential threshold to modulate the excitability of sensory neurons (in particular, non-peptidergic ones). In agreement, overexpression of the channel using viral vectors decreased the excitability of sensory neurons and diminished pain sensitivity in animal models [41,43,44] (Figure 1).

Removal of TRESK, rather than producing a general increase in excitability, has a selective effect on certain types of fibers/sensory neuron subtypes, thus affecting specific sensory modalities. Similar modulation of pain sensitivity to distinct stimuli has been encountered for other channels of the family. For instance, genetic ablation of TREK-1 modifies mechanical, heat and cold pain perception [45] but TRAAK deletion only enhances heat and mechanical sensitivity, while cold sensitivity remains unaffected [46]. Interestingly, if both channels are eliminated (TREK-1/TRAAK double KO mice), mice show an enhanced cold sensitivity compared to TREK-1^-/-^. Besides, deletion of TREK-2 only enhances thermal sensitivity to non-aversive warm conditions without affecting cold perception [47]. Knocking out TRESK produces a significant enhancement of mechanical and cold sensitivity, while sensitivity to heat remains largely unaffected or the effects are much smaller [23,25,39]. This seems to be related to a percentage of peripheral C-fibers activated by lower mechanical thresholds or in a warmer range of temperature, compared to fibers from WT mice [25]. These results suggest that in normal conditions, TRESK prevents the activation of nociceptive fibers by innocuous or low intensity stimuli, therefore only the high intensity ones can depolarize peripheral nerve endings sufficiently to produce neuronal firing. This seems restricted to certain modalities, but not others, such as heat sensing fibers in which excitability is probably regulated by other ionic mechanisms.

In addition to the downregulation of TRESK after axotomy, similar effects have been found in other models of persistent pain. During cutaneous inflammatory pain, a decrease in TRESK mRNA has been reported although the effects were rather small and in a limited number of animals [21]. In TRESK KO mice, an inflammatory pain model produced by injection of Complete Freund’s Adjuvant (CFA) in the hind paw did not show significant differences compared to WT mice, neither in the development of mechanical nor thermal hypersensitivity, except for the initially enhanced mechanical sensitivity present in KO animals [25]. Despite the data is not conclusive, it seems that the involvement of the channel in the development of inflammatory pain is rather limited. Models of neuropathic pain (complete axotomy, spared nerve injury or spinal nerve ligation) have shown a marked reduction of TRESK expression in the DRGs, which likely contributes to the enhanced mechanical and thermal hypersensitivity [18,40,44,48]. Recovery of TRESK expression levels by viral transfection partially reversed mechanical allodynia [40,44]. A similar TRESK downregulation in DRGs has been reported in a model of cancer-associated pain and overexpression of the channel suppresses tumor-induced neuronal hyperexcitability and pain hypersensitivity in this bone metastasis model [41]. In a nerve-cuffing induced neuropathic pain model, we did not find significant differences in the development of mechanical and thermal hypersensitivity between WT and TRESK KO animals, having a similar effect in both genotypes. Because KO mice had a lower mechanical threshold to begin with, animals reached lower mechanical thresholds 5 to 7 days post-injury, but mechanical hypersensitivity was undistinguishable between groups at later stages (14 and 21 days). Radiant heat sensitivity showed a similar level of hyperalgesia after neuropathy, in agreement with the lack of implication of TRESK in thermal perception [25]. Interestingly, TRESK KO mice presented cold allodynia, similar to that induced by oxaliplatin treatment in WT mice. In this sense, oxaliplatin treatment did not further enhance the cold hypersensitivity achieved by TRESK ablation [25]. Whether TRESK ablation produces other cellular changes driving to hypersensitivity of cold-sensing nociceptors remains to be elucidated.

### 3.2. Calcineurin Modulation of TRESK and Pain Sensitivity

The regulation of TRESK by calcineurin-mediated dephosphorilation [13], a unique feature among other K_2P_ channels, it is thought to contribute to TRESK function. Neurotransmitters or other compounds acting on membrane receptors that finally increase cytosolic calcium, specially Gαq-coupled receptors, will enhance TRESK current, thus opposing neuronal depolarization by these stimuli [13,36,49,50]. According to several studies, TRESK remains constitutively phosphorylated (at residues S252, S262, S264, and S267 in the human channel; Figure 2) in resting conditions and calcium binding to calcineurin phosphatase rapidly activates the channel by dephosphorylation. If phosphatase activity is discontinued by recovery of low calcium levels, the channel will be rephosphorilated by kinases and will return to the resting state. In this effect, protein kinase A (PKA) and microtubule affinity-regulating (MARK) kinases seem to play a significant role [13,50,51,52] (Figure 2). Besides, protein kinase C (PKC) indirectly activates the human channel, as treatment with phorbol 12-myristate-13-acetate (PMA) enhanced its current in *Xenopus laevis* oocytes [53]. In contrast, the mouse channel was reported to be insensitive to PMA [12]. Although the intermediate signaling pathway was not initially described, novel-type PKC isoforms (nPKC) seem to be involved (rather than conventional PKC isoforms). In any case, the effect was independent of direct PKC-mediated phosphorilation of the channel since mutation of putative PKC phosphorilation sites did not abolish the PMA effect [53]. Confirmation of this effect was later reported and the likely mechanism of action involves the nPKC-dependent inhibition of the kinase responsible for the rephosphorylation of the channel at S264 [54].

It has been proposed that calcineurin-inhibitor pain syndrome (CIPS) that occurs as a result of the phosphatase activity inhibition by immunosuppressive drugs, such as tacrolimus (FK506) or cyclosporin A, could be due to impaired modulation of TRESK current. This would increase neuronal excitability to enhance pain sensitivity [55]. This hypothesis has never been confirmed by detailed studies, but an interesting report has given some insights. Using a model of pain hypersensitivity in bone cancer–bearing rats, the authors show that intrathecal injection of calcineurin enhances TRESK mRNA and protein expression, as well as TRESK-mediated current [41]. These effects are correlated with both, a decrease in the excitability of DRG sensory neurons and an amelioration of mechanical and thermal hypersensitivity. Similar effects are found when doing a pretreatment with a NFAT inhibitor peptide, as NFAT (nuclear factor of activated T cells) is involved in the regulation of TRESK expression downstream of calcineurin activation. In contrast, exogenous application of tacrolimus or a calcineurin siRNA to cultured or in vivo DRGs increases TRESK phosphorylation and produced opposite effects (increase) on excitability and pain sensitivity. Similarly, interfering peptides that inhibit TRESK dephosphorilation render comparable effects on excitability. This data indicates that calcineurin inhibition deactivates TRESK channels through the suppression of TRESK dephosphorylation but also reduces TRESK protein and mRNA abundance. The study supports the hypothesis proposed for CIPS, although other effects of calcineurin inhibition apart from the regulation of TRESK should also be taken into consideration.

### 3.3. TRESK and Migraine

TRESK was involved in migraine pathophysiology in a study aimed to identify mutated brain-expressed ion channel genes in migraine patients [20]. A frameshift mutation in the KCNK18 gene (F139WfsX24) was present in several members of a family suffering from familial migraine with aura, while was absent in other family relatives that did not suffer from migraine pain. This mutation produces a truncated and non-functional TRESK channel (from 384 to 162 aa) with dominant-negative effects that can suppress wild-type channel function. The truncated channel includes the intact cytosolic N terminus, the first transmembrane region, the first extracellular loop and an aberrant sequence of 24 aa at the C terminus [20,56]. Expression of this mutated channel in HEK293 cells confirmed the dominant-negative effects observed in *Xenopus* oocytes and highlighted that not only the TRESK current is diminished (heterodimerization of wild-type and mutant TRESK subunits) but also the plasma membrane levels of the channel. When expressed in trigeminal sensory neurons, the mutated channel produced an increase in membrane resistance, a lower current threshold for action potential firing and an enhanced excitability of trigeminal sensory neurons.

Surprisingly, other mutations that inactivate TRESK were not correlated with migraine [57,58]. A number of KCNK18 missense variants (R10G, A34V, C110R, S231P, A233V) were also identified in unrelated sporadic migraine and control cohorts [20], as well as in Italian migraine patients [59]. Some variants (R10G, S231P, and A233V) did not appear to significantly affect TRESK current, [57] but, unexpectedly, the C110R variant, despite producing a complete loss of TRESK function, did not show an association with the disease, as it was expressed both in control and sporadic migraine cohorts. A similar outcome was reported for the A34V variant, although this mutant can still form functional channels. Furthermore, the C110R variant expressed in trigeminal neurons did not produce a significant increase in excitability [58]. These studies suggested that the presence of a single non-functional variant of TRESK alone is probably not sufficient to develop migraine.

These inconsistencies have been recently resolved in a study reporting that the TRESK mutation F139WfsX24 produces a nonfunctional channel (MT1) and a second protein fragment (MT2) that co-assembles with and inhibits TREK1 and TREK2 channels [60]. First, the study shows that TRESK can heteromerize with TREK1 or TREK2 subunits to form heteromeric channels. Second and more important, this mutation but not others (e.g., C110R), produces a second protein fragment due to the generation of a new translation initiation site (a mechanism termed frameshift mutation-induced alternative translation initiation) This second protein MT2 exerts a dominant negative effect on heteromeric channels, thus achieving a combined inhibition of TRESK, TREK1 and TREK2 channels. An increase in neuronal excitability is observed when MT2 is expressed in trigeminal neurons but not when only MT1 (truncated TRESK channel) is expressed. Finally, viral transfection of MT2 into the rat trigeminal ganglion produces a migraine-like phenotype, where an increase in facial mechanical sensitivity is observed over several days [60]. Interestingly, the facial mechanical allodynia elicited by injection of MT2, cannot be further enhanced by ISDN (isosorbide dinitrate) injection, a well-known model of migraine-like mechanical hypersensitivity. Database mining for other TRESK mutations found another described variant (Y121LfsX44) that again produces a second protein by the same mechanism. This mutation was also shown to produce a second protein that can interact with TREK channels. Importantly, this mutant has been associated with a migraine phenotype in the ClinVar database (www.ncbi.nlm.nih.gov/clinvar/).

In a recent study, human nociceptors with the F139WfsX24 mutation were generated from induced pluripotent stem cells from migraine patients [61]. Non-peptidergic nociceptors containing the mutation showed an increased excitability compared to these obtained from healthy control subjects. Correction of the mutation by CRISPR-Cas9 technology returned TRESK expression levels and neuronal excitability to normal values comparable to controls. Finally, the authors showed the therapeutic possibilities of enhancing TRESK activity. Cloxyquin, a TRESK activator that will be discussed in the next chapter, was able to reduce spontaneous firing in iPSC-derived nociceptors carrying the F1449V gain-of-function mutation in Nav1.7, that causes erythromelalgia [61]. In addition, in a rodent model of migraine generated by administration of nitroglycerine, cloxyquin treatment was able to reduce mechanical and thermal hypersensitivity, highlighting the potential of TRESK as an analgesic drug target. Recently, a new missense mutation (W101R) was identified in a male individual having intellectual disability and migraine with brainstem aura [62]. This mutation markedly reduces TRESK current amplitude, although the channel retains its K^+^ selectivity and the modulation by calcineurin. Interestingly, and in addition to the known association to migraine, the mutation links the channel to intellectual disability pathogenesis, a finding that remains to be confirmed by future studies.

### 3.4. TRESK and Mechanosensation

From the observation of tingling and numbing sensations produced by Szechuan peppercorns, Bautista et al. identified the natural compound hydroxy-α-sanshool as the responsible for activation of sensory neurons producing these effects [63]. Sanshool activates sensory neurons by depolarizing the cellular membrane through inhibition of the K_2P_ background channels TRESK, TASK1, and TASK3. It was proposed that the tactile component of these sensations might result from the activation of large-diameter, touch-sensitive fibers, whereas the pungent or irritant qualities may involve excitation of non-peptidergic, capsaicin-sensitive small-diameter nociceptive fibers. These studies suggest that tingling paresthesia elicited by sanshool is mediated by activation of mechanosensitive somatosensory neurons through inhibition of K_2P_ channels, in particular TRESK channels. In turn, numbing reported by sanshool might be mediated by desensitization of activated neurons. In fact, using the skin-nerve preparation, it has been shown that sanshool activates all ultrasensitive D-hair fibers and, to a lesser extent, some populations of pressure-sensitive Aβ fibers and low conduction velocity C fibers [64]. Later, it was described that sanshool also inhibits Na_v_1.3, Na_v_1.7, and Na_v_1.8, and the subsequent blocking effect on action potential firing is thought to mediate its analgesic effects by silencing Aδ mechanonociceptors [65]. A sanshool synthetic derivative, isobutylalkenyl amide (IBA) also induces tingling sensations in humans [66]. In rats, IBA inhibits TRESK current, activates nociceptive C-type fibers and induces nocifensive behaviors and mechanical allodynia [18,67]. Similar to sanshool or IBA, dermal exposure to pyrethroid insecticides causes paresthesias (tingling, itching, burning, and stinging sensations) due to their ability to activate sensory neurons. As the former compounds, pyrethroid insecticides inhibit TRESK (as well as other K_2P_ channels) and activate peripheral Aβ-, Aδ-, and C-fibers, increasing mechanical sensitivity when injected subcutaneously [68].

Another evidence of TRESK’s involvement in the function of mechanosensory neurons was obtained by stimulation of cultured DRG neurons with radial stretch, which activates nociceptors and mechanoreceptors [69]. Using hydroxy-α-sanshool and different TRP activators, two populations of stretch-sensitive neurons were described: low threshold mechanoreceptors/proprioceptors and non-peptidergic nociceptors. Both these populations of stretch-sensitive neurons express TRESK channels [69]. It has been shown that, in contrast to mechanically-gated channels, direct mechanical stimulation of the cell membrane does not activate or inhibit TRESK [70]. Nevertheless, changes in membrane tension modulates channel activity [37]. TRESK current is enhanced by cell swelling and inhibited by cell shrinkage, indicating that changes in membrane curvature/shape/tension modify channel activity/opening. The use of membrane crenators or cup-former substances (e.g., volatile anesthetics), which preferentially insert into the external or internal leaflet of the bilayer, respectively, and change membrane tension, corroborate these findings. In fact, the potent activation of TRESK by volatile anesthetics, such as isoflurane or halothane, could be mediated by modulation of the channel through membrane lipids.

In summary, a relationship between TRESK and mechanical sensitivity has been well demonstrated. The increase in mechanical sensitivity observed in different studies (knocking out TRESK in mice; mutations that knockdown the channel; pharmacological TRESK inhibition) suggests that TRESK is involved in the transduction of mechanical stimuli, not as a direct sensor but by regulating the excitability of sensory neurons involved in mechanodetection. Whether modulation of TRESK by membrane tension also contributes to the detection of mechanical stimuli in peripheral nerve endings remains to be studied in detail.

## 4. TRESK Modulation and Pharmacology: Therapeutics

Basic biophysical properties, as well as modulation of TRESK activity by different intracellular kinases and phosphatases or interaction with several proteins, has been nicely reviewed [2,71,72]. As mentioned before, several compounds inhibiting the channel have been described, including sanshool, IBA, pyrethroids, quinine, quinidine, arachidonic acid, amide local anesthetics (e.g., bupivacaine) or different ions (Ba^2+^, Hg^2+^). For therapeutic purposes, TRESK activating compounds (openers) have more interest due to their action diminishing neuronal activation and producing analgesic effects. Besides the activating effects of volatile anesthetics on TRESK and other K_2P_ channels, only a few compounds have been described as TRESK activators. Flufenamic acid, BL-1249, and its synthetic derivatives are TRESK openers that seem to act through a stabilization of an open state of the channel [73]. Using a baculovirus TRESK FluxOR cell-based assay, Wright et al. identified cloxyquin (5-chloroquinolin-8-ol), an old drug used to treat intestinal infections due to its anti-amoebic, anti-bacterial and anti-fungal properties [74]. Cloxyquin analogues generated accordingly to the structure-activity relationship also behaved as channel activators. Interestingly, the drug seems to possess some selectivity as it did not affect other potassium channels as ROMK (Kir1.1) or hERG (human Ether-à-go-go-Related Gene, Kv11.1) [74]. In addition, it is selective for TRESK among other K_2P_s [75]. Cloxyquin activates TRESK in its resting state, by a mechanism independent of Ca^2+^/calcineurin-mediated dephosphorilation. In fact, cloxyquin did not further increase TRESK after activation of the channel by receptor-mediated calcium increase or in mutant channels mimicking the dephosphorylated state [75]. When tested in DRG sensory neurons, cloxyquin also increased the background K^+^ current in a subset of neurons, indicating that it is able to activate native channels [68,75]. As mentioned earlier, cloxyquin reduced spontaneous firing in iPSC-derived nociceptors carrying the F1449V gain-of-function mutation in Nav1.7, but, more importantly, this drug reduced mechanical and thermal hypersensitivity in a mouse model of migraine by administration of nitroglycerine [61]. The identification of cloxyquin and its in vitro and in vivo effects reinforce the therapeutic potential of targeting TRESK for analgesic purposes and opens the door to find new and selective compounds to activate the channel for the treatment of migraine or other pain-related diseases.

In the search and development of TRESK channel openers or inhibitors, species differences need to be taken into account. Despite the similarities between the human and rodent (mouse or rat) channels, some pharmacological differences have been described. TRESK inhibition by local anesthetics (bupivacaine and lidocaine) has about 10-fold greater potency on rodent TRESK than on the human ortholog. In contrast, activation by volatile anesthetics is more prominent on the human channel [15,76]. Differences also exist on the sensitivity to other stimuli, such as pH or Zn^2+^, being human TRESK almost insensitive but not mouse and rat orthologs. Interestingly, cloxyquin similarly activates the human and rodent channel, thus not showing significant pharmacological differences.

Some differences have also been described regarding the intracellular regulation of the channel. Human TRESK interact with anionic membrane phospholipids and is regulated by PIP_2_ generated by activation of GPCRs, whilst, in rodent orthologs of the channel, this regulation is not present [77]. Significant differences are also present in other species (e.g., zebrafish), where TRESK channels are not regulated by Ca^2+^/calcineurin [78]. These species differences must be taken into consideration when extrapolating pharmacological results from rodent or other experimental animal models to human physiology, as well as during drug development.

## 5. Conclusions

As described, several studies to date have highlighted the role of TRESK in regulating the excitability of specific subtypes of sensory neurons. Targeting the channel seems an interesting therapeutic approach to reduce nociceptor activation and decrease pain. Some recent studies have pointed out the involvement of the channel in the regulation of the sensitivity to specific types of stimuli but not others. In addition, the involvement of the channel seems more prominent in certain alterations of the sensitivity (mechanical or cold allodynia) rather than other painful conditions (inflammatory pain). The contribution of the channel to specific sensory modalities and on the pathophysiology of different pain-related diseases remains to be fully established, but it seems plausible that enhancing the channel function could ameliorate mechanical or cold allodynia experimented by many patients suffering from diverse medical conditions (e.g., migraine) or pharmacological treatments (e.g., oxaliplatin). The fact that the channel expression is restricted to certain types of neurons, mainly sensory neurons, makes it an attractive target for a selective therapeutic intervention. In addition, strategies to increase or recover channel expression after injury could be beneficial for certain neuropathic painful conditions. Development of specific activators of the channel is desired to expand the current available analgesic agents.

## Figures and Tables

**Figure 1 ijms-21-05206-f001:**
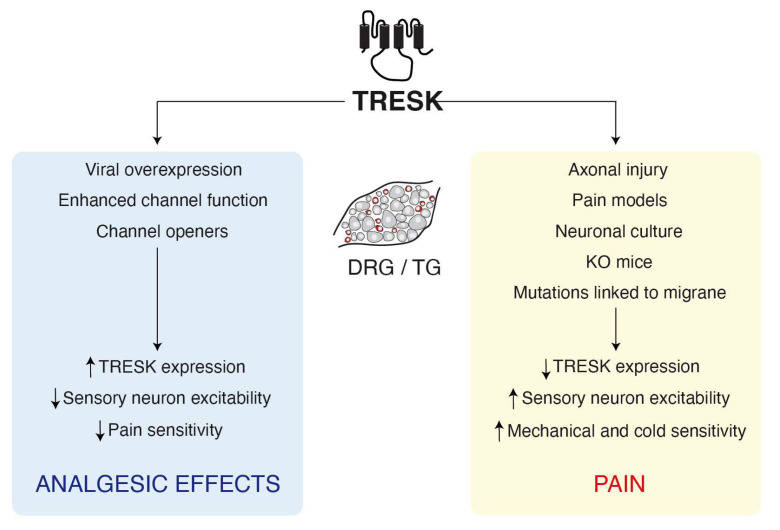
TRESK regulation is linked to pain or analgesic effects. Axonal and neuronal injuries or TRESK mutations conduct to a decrease in channel function that leads to an increase in pain sensitivity. Enhancement of TRESK expression or channel openers decreases the excitability of sensory neurons producing analgesic effects.

**Figure 2 ijms-21-05206-f002:**
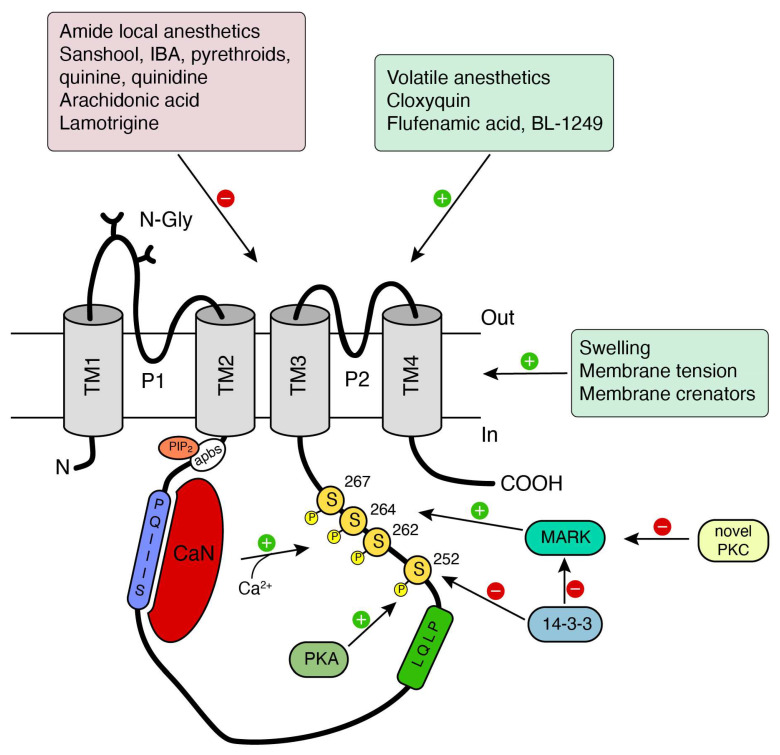
Regulation of TRESK by extracellular stimuli or by intracellular pathways and interacting proteins. Schematic model of the human TRESK channel in the resting state. Known activators and inhibitors/blockers of the channel are indicated on top, as well as stimuli that modulates channel function through the membrane. Regulatory and interaction sites with membrane lipids and PIP_2_ (apbs) or calcineurin (PQIIIS and LQLP) are shown. Calcineurin interacts with the LQLP motif upon calcium binding to dephosphorilate serines S262, S264, and S267. Other regulatory proteins (PKA, 14-3-3 and microtubule affinity-regulating kinases (MARK) and sites of action are indicated.

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
