# Peer review of "The Background K+ Channel TRESK in Sensory Physiology and Pain"

_ijms, 2020, doi:10.3390/ijms21155206_

Round 1

Reviewer 1 Report

July 17, 2020

TRESK, the most recently identified member of the two pore-domain potassium channel family, has been investigated for nearly two decades. TRESK and other K2ps (TWIK1, TASK1, TASK2, TREK1, TREK2, and TRAAK) together have been simply reviewed as pain related channels recently. In this manuscript, the authors point out and emphasize that TRESK possibly could be a promising target in sensory neuron for pain therapy. The authors correctly interpreted and presented the relevant results published in this field.

I only have three recommendations:

Line 14, please rewrite the sentence of “Therefore, the channel has been involved in pain sensitivity.” I recommend to use the following sentence: “These findings suggest TRESK could be involved in pain sensitivity.”

Line 82, please add “in 2003” at the end of the first sentence to indicate clearly when the gene was cloned.

Line 55-58, please add the channel name (K2P18) and gene name (KCNK18) for TRESK in the beginning of these sentences.

Author Response

The authors would like to thank the reviewer for revising the manuscript and providing positive comments. The suggested changes have been introduced in the revised version of the manuscript. 

Line 14, please rewrite the sentence of “Therefore, the channel has been involved in pain sensitivity.” I recommend to use the following sentence: “These findings suggest TRESK could be involved in pain sensitivity.”

Corrected.

Line 82, please add “in 2003” at the end of the first sentence to indicate clearly when the gene was cloned.

Corrected.

Line 55-58, please add the channel name (K2P18) and gene name (KCNK18) for TRESK in the beginning of these sentences.

Added.

Reviewer 2 Report

This is a well-written focused review on the background K+ channel TRESK expressed primarily by primary sensory neurons. The authors summarize current knowledge on the genetics, molecular biology, expression, physiological function and pathobiological significance of this channel. The channel plays a significant role in the regulation of membrane excitability of primary afferent neurons. The role of this channel in the mechanisms of pain, especially phenomena associated with migraine is comprehensively highlighted by the authors. The possibility of a novel approach by modulating TRESK channel function to achieve antinociceptive effect has also been suggested.

Author Response

The authors would like to thank the reviewer for revising the manuscript and providing positive comments. The changes suggested by Reviewer1 have been introduced in the revised version of the manuscript.